# RETHINKING LLM PARAMETRIC KNOWLEDGE AS CONFIDENCE FOR EFFECTIVE AND EFFICIENT RAG

## ABSTRACT

Large Language Models (LLMs) tend to generate high-confidence hallucinations when faced with questions beyond their parametric knowledge scope. Retrieval-Augmented Generation (RAG) alleviates this by leveraging external knowledge, but challenges remain as to whether the retrieved context is useful (effective RAG) and whether to retrieve (efficient RAG) when answering specific-domain questions. This challenge underscores the importance of knowledge boundary awareness, which the current methods—relying on discrete labels or limited signals—fail to address adequately, as they overlook the rich information in LLMs' continuous internal hidden states. To this end, we propose a novel knowledge probing approach for effective and efficient RAG. First, we construct a confidence detection model based on LLMs' internal hidden states to quantify how retrieved contexts enhance the model's confidence. Then, we build a preference dataset with the confidence detection model to fine-tune a reranker, enabling it to prioritize contexts preferred by the downstream LLM. Additionally, we introduce CBDR, which adaptively triggers retrieval based on the LLM's initial confidence in the original question, reducing knowledge conflicts and improving efficiency. Experimental results show that significant improvements have been achieved in the accuracy of both context screening and end-to-end Retrieval-Augmented Generation (RAG) performance. Wherein, when dynamic retrieval is activated, the accuracy of the RAG system increases by 5.6 percentage points (pp); meanwhile, the retrieval cost is significantly reduced by 7.1 pp, thereby substantially enhancing the system's practical utility while maintaining competitive accuracy.

## 1 INTRODUCTION

The core efficiency bottlenecks of Retrieval-Augmented Generation (RAG) consistently revolve around two key issues: how to precisely select effective retrieval contexts and when to trigger retrieval. If retrieval contexts are irrelevant to the question, they will introduce knowledge conflicts and increase costs; if retrieval is forced when unnecessary, it will waste resources and reduce efficiency (Yoran et al. (2023); Fang et al. (2024)). Essentially, these problems stem from the "perceptual blind spot" of Large Language Models (LLMs) regarding their own knowledge boundaries—when faced with questions beyond the scope of their parametric knowledge, LLMs often generate hallucinations due to overconfidence (Ji et al. (2023); Martino et al. (2023)), failing both to judge "whether external knowledge is needed" and to identify "which external knowledge is truly useful," ultimately undermining the accuracy and practicality of RAG systems.

Existing studies have attempted to address this dilemma through "knowledge boundary awareness" but still exhibit limitations: Prompt-guided confidence estimation (Yin et al. (2023)) relies on manually designed templates, resulting in insufficient generalizability; multi-sample confidence aggregation (Brown et al. (2024)) is costly and ignores dynamic contextual influences; hidden-state-based methods (Su et al. (2024b); Ni et al. (2025)), while capturing continuous confidence signals, only stop at discrete labels outputs of "answerable/unanswerable" and fail to directly link confidence with "retrieval context selection."

To this end, this paper specifically proposes a model self-confidence-centric RAG framework to enhance RAG system efficiency: 1) Perceiving knowledge boundaries through confidence self-assessment: Inspired by Ni et al. (2025). A confidence detection model is trained to enable LLMs

to dynamically evaluate their confidence in answering original questions—low confidence triggers retrieval, while high confidence allows direct answer generation to reduce unnecessary operations; 2) Optimizing retrieval context reranking using confidence changes: Based on the magnitude of confidence improvement in LLMs when exposed to different retrieval contexts, a preference dataset is constructed to fine-tune the reranker, enabling it to prioritize contexts that "significantly enhance answer confidence," thus achieving direct translation from the model's intrinsic preferences to retrieval context reranking.

This logic can be intuitively understood through Figure 1: As illustrated in Figure 1a, this figure contrasts two architectures: one with a context similarity-based reranker and the other with a reranker based on the downstream LLM's confidence. Figure 1b further compares the differences between Contexts C1 and C2 in assisting the same LLM in completing question-answering tasks.

Based on the above ideas, the core technologies of this paper include: 1) Reranker fine-tuned with confidence signals: First, the confidence detection model parses the internal hidden states of LLMs to quantify the enhancement effect of different retrieval contexts on answer confidence (magnitude of confidence improvement); this quantitative signal is then used as supervision to fine-tune the reranker, enabling it to directly output retrieval contexts rankings consistent with the LLM's confidence preferences and prioritize contexts that significantly enhances answer reliability. 2) **C**onfidence-**B**ased **D**ynamic **R**etrieval (CBDR): Combined with the LLM's initial confidence in the original question, it adaptively decides whether to trigger retrieval, balancing accuracy and retrieval costs.

Experiments validate the effectiveness of this framework: a 5.6% improvement in end-to-end RAG accuracy and a 7.1% reduction in retrieval costs. The core contribution of this paper lies in the first-time deep integration of LLMs' confidence self-assessment with retrieval context reranking, providing a new approach for enhancing RAG system efficiency.

## 2 RELATED WORK

### 2.1 KNOWLEDGE BOUNDARY IN RAG SYSTEM

RAG's knowledge boundary is defined as the combined knowledge space of LLMs' internal parametric knowledge and external retrieved knowledge. Early RAG evaluations overemphasized retriever performance, neglecting potential conflicts between external and internal knowledge—which lead to low-confidence errors (Yoran et al. (2023); Fang et al. (2024); Cuconasu et al. (2024)).

Subsequent research shifted to coordinating these dual knowledge sources to delineate RAG's effective boundary. Works like (Marina et al. (2025); Yao et al. (2024)) analyze LLMs' internal states to detect uncertainty and dynamically trigger retrieval. DRAGIN (Su et al. (2024a)) dynamically retrieves information by assessing the importance and uncertainty of generated tokens during inference. CTRLA (Liu et al. (2024)) quantifies confidence by computing the projection of the current query onto the LLM's confidence representation, thereby dynamically triggering retrieval. Adaptive-RAG (Jeong et al. (2024)) employs a lightweight model to estimate question complexity and select an appropriate retrieval strategy. Like CBDR, Probing-RAG (Baek et al. (2025)) trains a small model to inspect the internal states of the target LLM but does not exploit the resulting state discrepancies to inform retrieval preferences. Parenting (Xu et al. (2025)) automatically defines knowledge boundaries by quantifying the relative importance of two capabilities—adherence and robustness—through parameter analysis. DTA framework (Sun et al. (2025)) formally proposes RAG's knowledge boundary, categorizing queries into four quadrants based on LLM's parametric boundary $KB_p$ and retriever's retrieved boundary $KB_r$ to define the system's holistic effective boundary.

### 2.2 PREFERENCE ALIGNMENT IN RAG SYSTEM

To improve LLMs' utilization of external knowledge, aligning retriever-LLM preferences in RAG is critical. Existing works use diverse preference signals: RE-PLUG (Shi et al. (2023)): LLM's correct answer probability to identify critical contexts; RRR (Cong et al. (2024)): Overall quality of LLM-generated responses; DPA-RAG (Dong et al. (2025a)): Bidirectional alignment to mitigate component preference conflicts; RADIO (Jia et al. (2024)): Rationale correctness as indicators,

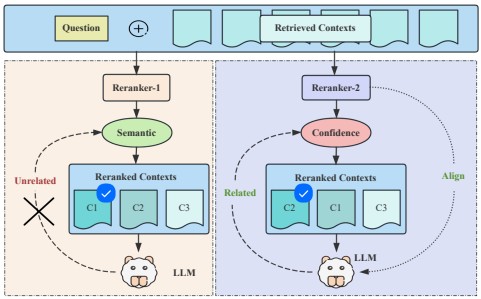

| Query: | who played karen in married to the mob? |
|---|---|
| **Reference:** | Nancy Travis |
| **C1:** | **Married to the Mob** is a 1988 American comedy film directed by Jonathan Demme, starring Michelle Pfeiffer and Matthew Modine. |
| **C2:** | ..., who gets violently dispatched by **Mob** boss Tony "The Tiger" Russo (Dean Stockwell) when he is discovered in a compromising situation with the latter's mistress **Karen (Nancy Travis).** ... |
| **A1:** | Based on the provided context, I can answer the question. According to the context, **Michelle Pfeiffer played a role in the film "Married to the Mob"**, but ... |
| **A2:** | Based on the provided context, **Nancy Travis played the role of Karen**, Tony "The Tiger" Russo's mistress. |

(a) Comparison of Reranking Strategies          (b) Performance Evaluation

Figure 1: Figure 1a contrasts two RAG reranking strategies: a conventional context-similarity-based reranker and one leveraging the LLM's intrinsic preference (reranking contexts by changes in the LLM's confidence). Figure 1b provides a concrete example: it compares the effectiveness of contexts reranked by Reranker-1 (similarity-based) and Reranker-2 (LLM-aligned).

fine-tuning rerankers to reconcile retriever-LLM discrepancies; SEAKR (Yao et al. (2024)): Multi-round query sampling, using LLMs' last-layer hidden states (at $$) to compute Gram matrices (quantifying uncertainty) for reranker optimization.

This paper's core innovation is a novel preference metric: *confidence shift*, defined as LLMs' internal hidden state changes before/after exposure to external knowledge. Used to fine-tune rerankers, it effectively filters post-retrieval contexts. Compared to SEAKR (Yao et al. (2024)) (which also uses hidden states but requires multi-round sampling), our confidence shift detection relies on a single forward pass—significantly reducing computational/temporal overhead, a key advantage for low-latency real-time scenarios.

## 3 METHOD

This section outlines our core methodologies: leveraging LLM internal hidden states to assess response confidence, constructing a preference dataset from these states to fine-tune a Reranker, and proposing CBDR to optimize retrieval in RAG system. Relevant prompts are in Appendix B.

### 3.1 INTERNAL STATE DETECTION

Recent studies show that LLMs' internal hidden states contain richer information (stronger latent reasoning, self-awareness) than their final output token (Skean et al. (2025); Zhang et al. (2025); Azaria & Mitchell (2023)) and that LLMs can perceive their knowledge boundary before response generation (Ni et al. (2025)), laying the foundation for confidence estimation via these internal hidden states.

### 3.1.1 CONFIDENCE ESTIMATION VIA INTERNAL HIDDEN STATES

Specifically, the workflow for self-confidence detection based on the internal hidden states of LLM is as follows: For a given target LLM M and a question Q, the model generates internal hidden state representations during inference, denoted as $H_{M,Q}$. Compared to the final token output, this state encapsulates more comprehensive information. Our confidence estimation process is defined as:

$$C_{M,Q} = E(H_{M,Q}) \tag{1}$$

As illustrated in left side of Figure 2, where E denotes the confidence detection model, and $C_{M,Q}$ is a binary classification label: $C_{M,Q} = 1$ indicates that LLM M is confident in correctly answering question q, whereas $C_{M,Q} = 0$ signifies that the LLM M perceives itself as incapable of responding accurately. Drawing on (Ni et al. (2025)) and related prior work, we select the internal hidden state vector at Mid_Layer (Layer/2) of LLM M before generating the first answer token (Pre-Token) as $H_{M,Q}$.

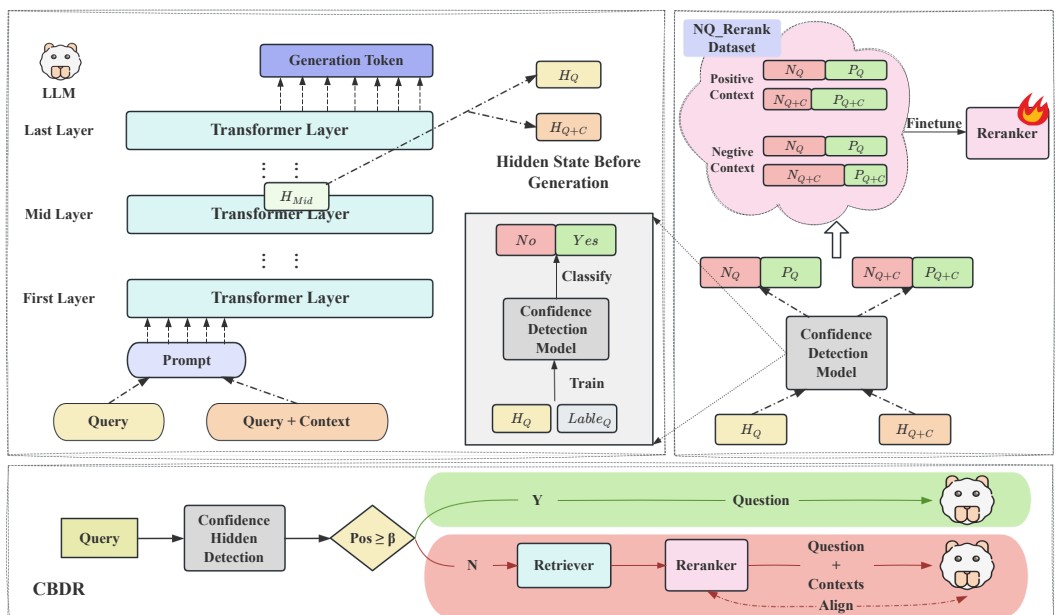

Figure 2: The complete process of aligning the Reranker with the target LLM involves constructing a preference dataset, NQ_Rerank, by comparing the variations in the LLM's confidence when answering a Question under different contexts. This dataset is then used to fine-tune the Reranker model, aligning it with the target LLM's intrinsic preferences. The lower part of the figure depicts the workflow of the CBDR.

The training data for model E is obtained by guiding LLM M to process questions from the NQ dataset (Kwiatkowski et al. (2019)). We collect the internal hidden state $H_{M,Q}$ during inference and determine the correctness of the LLM M's response based on the ground-truth answer to question Q, thereby constructing binary training samples $(H_{M,Q}, Label_Q)$. Here, $Label_Q = 1$ indicates that the model answers question Q correctly, while $Label_Q = 0$ denotes an incorrect response. The training methodology for model E follows the approach described in (Ni et al. (2025)). We performed data cleaning on the training dataset NQ Kwiatkowski et al. (2019) and analyzed the impact of different prompt designs on model reasoning.

## 3.2 Preference Dataset

### 3.2.1 Preference Definition

This study focuses on the post-retrieval processing stage within RAG system, with the aim of exploring how to rerank the retrieved contexts to maximize RAG system's utility in enhancing the answer reasoning capabilities of downstream LLM.

Conventional Reranker are typically trained on datasets constructed based on semantic similarity between a question and contexts, and compute relevance scores by capturing complex semantic interactions through interactive encoding. While such general-purpose methods ensure model transferability and compatibility with diverse LLMs, they often fail to adequately incorporate the preferences of specific downstream LLM, thereby limiting the full potential of RAG system.

$$Conf(H_{M,Q}) = P(Label = 1 \mid E(H_{M,Q})) = Softmax(Z_1) = \frac{e^{z_1}}{e^{z_0} + e^{z_1}} \qquad (2)$$

As illustrated in Figure 2, this paper defines the following preference criterion: a context $C$ is considered to exhibit a positive preference for the target LLM $M$ in answering question $Q$ if and only if it provides effective informational enhancement, satisfying the condition $Conf(H_{M,Q+C}) > Conf(H_{M,Q})$. Conversely, if it leads to a decrease in LLM's confidence $Conf(H_{M,Q+C}) < Conf(H_{M,Q})$, the context $C$ is regarded as having a negative preference. As shown in Equation 2, the output of the Conf(·) function is defined as the probability of the $Label = 1$ assigned by model

E. A softmax layer is appended to the final layer of model E to produce this probabilistic output. Relevant examples can be found in Appendix D.

### 3.2.2 DATASET CONSTRUCTION

We preprocess the NQ dataset (Kwiatkowski et al. (2019)) to obtain a series of $(Query, Contexts)$ tuple samples. For each sample, we record the internal hidden state at Mid_Layer when the target LLM M generates its first token under the following two scenarios: (1) The state $H_{M,Q}$ when only the query $Q$ is provided; (2) The state $H_{M,Q+C_i}$ when both the query $Q$ and a context $C_i$ are provided (Where i iterates over the Contexts).

This yields a sequence of internal hidden states:

$$[H_{M,Q}, \ H_{M,Q+C_1}, \ H_{M,Q+C_2}, \ ..., \ H_{M,Q+C_i}] \tag{3}$$

This sequence of states is then fed into the confidence detection model E to obtain the probability value for the $Label = 1$ output by the softmax layer, resulting in a probability sequence:

$$[Conf(H_{M,Q}), \ Conf(H_{M,Q+C_1}), \ ..., \ Conf(H_{M,Q+C_i})] \tag{4}$$

The enhancement effect of each context $C_i$ on LLM M's response to question $Q$ is determined by comparing the change in model confidence after incorporating the context $C_i$:

$$Inc(Q, C_i) = Conf(H_{M,Q+C_i}) - Conf(H_{M,Q}) \tag{5}$$

If $Inc(Q, C_i) > 0$, the sample is labeled as a positive preference sample. If $Inc(Q, C_i) < 0$, it is labeled as a negative preference sample.

For each $(Query, Contexts)$ sample, all context $C_i$ are ranked according to $Inc(Q, C_i)$. The Top-K(K = 5) contexts with the highest increase are selected as positive examples, and the Top-K with the largest decrease are taken as negative examples. As illustrated in right side of Figure 2, this process constructs the final preference dataset, denoted as NQ_Rerank. Relevant details can be found in Appendix E.

### 3.3 RERANKER FINE-TUNING

To enhance the ability of the Reranker to identify the utility of contexts for the target LLM, we performed supervised fine-tuning on a base Reranker using the constructed preference dataset NQ_Rerank. During fine-tuning, the InfoNCE (Noise Contrastive Estimation) loss function was employed as the optimization objective:

$$f(Q, C) = exp(\phi(Q, C)/\tau) \tag{6}$$

$$L = -log \frac{f(Q, C^+)}{f(Q, C^+) + \sum_{i=1}^{N} f(Q, C_i^-)} \tag{7}$$

Where: $f(Q, C)$ denotes the relevance score between question $Q$ and context $C$ computed by the Reranker; $C^+$ represents the positive context; $C^-$ denotes the negative context; $\tau$ is the temperature parameter. This loss function forces the model to increase the score margin between the positive context $C^+$ and a set of negative contexts $\{C^-\}$, thereby learning a ranking criterion consistent with the target LLM's preferences.

### 3.4 CONFIDENCE-BASED DYNAMIC RETRIEVAL

While the fine-tuned Reranker has aligned well with the target LLM's preferences and effectively prioritizes beneficial contexts, it still has two key limitations: namely, the post-retrieval Top-k results may contain misleading context that conflicts with the LLM's internal parameters; and for questions for which the LLM is overconfident, the retrieval process can be skipped altogether to avoid redundant computational overhead.

To mitigate these issues and enhance the efficiency and reliability of the RAG system, we propose CBDR. The workflow of this strategy is illustrated at the bottom of Figure 2: (1) If the target LLM exhibits high confidence in responding to the current query $Q$ that $Conf(H_{M,Q}) > \beta$, where $\beta$ is a predefined threshold, the retrieval and reranking steps are skipped, and the LLM generates the

Table 1: Performance Comparison of Different Rerankers on the NQ_Rerank Test Set.

| Reranker | Params | Top-1 | | | Top-3 | | | Top-5 | | |
|---|---|---|---|---|---|---|---|---|---|---|
| | | Precision | Recall | MRR | Precision | Recall | MRR | Precision | Recall | MRR |
| gte_passage-ranking_multilingual-base | 304M | 85.52 | 29.47 | 85.52 | 71.45 | 62.66 | 90.37 | 60.98 | 82.53 | 90.99 |
| Qwen3-Reranker | 4B | 81.74 | 27.62 | 81.74 | 70.92 | 62.33 | 88.15 | 61.71 | 83.53 | 88.93 |
| Qwen3-Reranker | 8B | 87.25 | 30.47 | 87.25 | 74.35 | 65.15 | 91.65 | 64.22 | 86.42 | 92.19 |
| bge-reranker-v2-m3 | 568M | 86.01 | 29.45 | 86.01 | 72.62 | 63.61 | 90.47 | 62.40 | 84.01 | 91.07 |
| bge-reranker-v2-m3-ft (Ours) | 568M | **91.20** | **32.01** | **91.20** | **76.98** | **67.14** | **94.40** | **65.64** | **87.97** | **94.72** |

answer directly. (2) If the confidence score falls below the threshold $Conf(H_{M,Q}) < \beta$, the full retrieval process is initiated: the Retriever fetches a set of contexts, which are reranked by the fine-tuned Reranker, and the Top-K contexts are fed into the LLM along with the query for reasoning.

To mitigate these issues and enhance the efficiency and reliability of the RAG system, we propose CBDR. The workflow of this strategy is illustrated at the bottom of Figure 2: (1) If the target LLM exhibits high confidence in responding to the current query $Q$ that $Conf(H_{M,Q}) > \beta$, where $\beta$ is a hyper-parameter, the retrieval and reranking steps are skipped, and the LLM generates the answer directly. (2) If the confidence score falls below the threshold $Conf(H_{M,Q}) < \beta$, the full retrieval process is initiated—with the fine-tuned Reranker involved.

This strategy aims to preserve answer quality while cutting redundant computation for high-confidence queries and avoiding interference from low-quality retrieval results for known questions. Its effectiveness is fully validated in Section 4.2.

## 4 EXPERIMENTS

### 4.1 EXPERIMENTAL SETUP

**Datasets.** We use two open-domain QA benchmarks: Natural Questions (NQ) Kwiatkowski et al. (2019) and HotpotQA Yang et al. (2018). NQ contains real Google queries with retrieved contexts and human-annotated answers; HotpotQA requires multi-hop reasoning. All training data in this work are from NQ, partitioned as follows: (1) *NQ_Confidence* for confidence detection model $E$: 1k/300/500 positive and negative samples for train/dev/test, labeled by the correctness of target LLM $M$'s answers; (2) *NQ_Rerank* for preference alignment: built on NQ-Retrieval[1], excluding samples without valid positive/negative contexts, yielding 7,622 training and 1,216 evaluation samples.

**LLMs.** We evaluate with Llama3-8B-Instruct Dubey et al. (2024) and Qwen2.5-7B-Instruct Team (2024). Llama3-8B-Instruct serves as the base LLM for reasoning (temperature = 1.0, greedy decoding).

**Baselines.** We compare five rerankers, all based on or compared against *bge-reranker-v2-m3*: (1) *gte_passage-ranking_multilingual-base* (Alibaba DAMO's strong multilingual reranker); (2) *Qwen3-Reranker-4B* and (3) *Qwen3-Reranker-8B* (Qwen-based rerankers of 4B/8B parameters); (4) *bge-reranker-v2-m3* (lightweight, efficient, multilingual); (5) *bge-reranker-v2-m3-ft (Ours)*: fine-tuned on NQ_Rerank to validate confidence-based preference alignment.

**Dynamic Retrieval Methods.** We compare with two strong dynamic retrieval baselines: (1) **DRA-GIN (Su et al. (2024a))**: triggers retrieval based on token-level uncertainty, importance, and relevance; (2) **CtrlA (Liu et al. (2024))**: adaptively balances internal/external knowledge via LLM state characterization and confidence monitoring using directional feature representations.

**Evaluation Metrics.** We report standard reranking metrics: $Precision@K$, $Recall@K$, and $MRR@K$. To ensure fairness, all rerankers receive the same retrieved context pool (retriever is excluded). Training details are in Appendix C.

---

[1]https://modelscope.cn/datasets/sentence-transformers/NQ-retrieval

Table 2: Accuracy of RAG Systems with Different Reranker and LLM Combinations. Reranker bge-reranker-v2-m3-ft is the reranker aligned with the confidence preferences of Llama3-8B-Instruct; bold indicates the optimal result, and underlined indicates the sub-optimal result.

| LLM | Reranker | Params | HotpotQA | | NQ | |
|---|---|---|---|---|---|---|
| | | | Top-1 | Top-3 | Top-1 | Top-3 |
| Qwen2.5-7B-Instruct | gte_passage-ranking_multilingual-base | 304M | 47.20 | 51.80 | 63.80 | 67.60 |
| | Qwen3-Reranker | 4B | 42.30 | 50.10 | 50.70 | 64.00 |
| | Qwen3-Reranker | 8B | 47.50 | 51.90 | 56.30 | 68.80 |
| | bge-reranker-v2-m3 | 568M | 47.20 | **53.30** | **64.20** | 69.70 |
| | bge-reranker-v2-m3-ft (Ours) | 568M | **48.70** | **53.30** | 63.40 | **69.90** |
| Llama3-8B-Instruct | gte_passage-ranking_multilingual-base | 304M | **48.80** | 50.20 | 60.10 | 60.70 |
| | Qwen3-Reranker | 4B | 40.70 | 48.00 | 49.70 | 62.30 |
| | Qwen3-Reranker | 8B | 48.40 | 50.10 | 55.20 | **68.80** |
| | bge-reranker-v2-m3 | 568M | 46.60 | 51.40 | 61.50 | 62.20 |
| | bge-reranker-v2-m3-ft (Ours) | 568M | 48.00 (+1.4) | **52.20** (+0.8) | **62.60** (+1.1) | 66.90 (+4.7) |

Table 3: Comparison of Different Dynamic Retrieval Methods. Bold indicates the optimal result, and underlined indicates the sub-optimal result. All methods use Llama3-8B-Instruct as the LLM. In the threshold, $h$ represents the hallucination score, $T$ represents the sensitivity of confidence monitoring, and $\beta$ indicates the confidence score.

| Method | Threshold | Reranker | RR (%) $\downarrow$ | Offline Cost | Online Cost | NQ | |
|---|---|---|---|---|---|---|---|
| | | | | | | Top-1 | Top-3 |
| DRAGIN (Su et al. (2024a)) | $h = 0.80$ | Bge-reranker-v2-m3 | 2.8 | No | High | 41.1 | 41.8 |
| | $h = 0.70$ | | 4.4 | | | 41.2 | 41.9 |
| | $h = 0.60$ | | 9.6 | | | 43.8 | 41.9 |
| | $h = 0.50$ | | 19.4 | | | 44.9 | 45.7 |
| CtrlA (Liu et al. (2024)) | $T = 0.00$ | Bge-reranker-v2-m3 | 53.4 | Low | High | 51.4 | 52.4 |
| | $T = 0.10$ | | 70.1 | | | 56.2 | 56.2 |
| CBDR (ours) | $\beta = 0.90$ | Bge-reranker-v2-m3-ft (ours) | 74.9 | Mid | Low | 60.6 | 64.2 |
| | $\beta = 0.95$ | | 83.3 | | | 62.4 | 66.1 |
| | $\beta = 0.98$ | | 92.9 | | | 61.7 | **67.8** |
| | $\beta = 1.00$ | | 100 | | | **62.6** | 66.9 |

## 4.2 MAIN RESULTS

**Reranker Performance.** We evaluate whether the fine-tuned reranker (bge-reranker-v2-m3-ft) better selects LLM-suitable contexts for Llama3-8B-Instruct on the NQ_Rerank test set. Each reranker ranks query-context pairs and returns Top-$K$ documents ($K \in \{1, 3, 5\}$), evaluated via $Precision@K$, $Recall@K$, and $MRR@K$. As shown in Table 1: (1) bge-reranker-v2-m3-ft (fine-tuned on NQ_Rerank) achieves the best performance across all $K$; (2) Gains are most pronounced at Top-1: it outperforms Qwen3-Reranker-8B by +3.95 pp in $Precision@1/MRR@1$ and +1.54 pp in $Recall@1$, and improves over the original bge-reranker-v2-m3 by +5.19 pp ($Precision@1/MRR@1$) and +2.56 pp ($Recall@1$).

**RAG System Accuracy.** To further verify whether a fine-tuned Reranker enhances Retrieval-Augmented Generation (RAG) system performance, we constructed multiple "Reranker + LLM" combined systems for comparative experiments. The setup was as follows: each Reranker reranked the query and its corresponding context documents, selected Top-K (K $\in$ \{1, 3\}) documents, and fed them to the downstream LLM for answer generation. System performance was ultimately evaluated by the precision of generated answers. As shown in Table 2, the experimental results yield two main findings: (1) When the downstream LLM was Llama3-8B-Instruct, the RAG system using bge-reranker-v2-m3-ft consistently achieved higher accuracy than that with the original bge-reranker-v2-m3, with a maximum improvement of +4.7 pp; this combination also reached optimal

| Question | Context | Context | Confidence | Answer |
|---|---|---|---|---|
| love yourself by justin bieber is about who | No | | 0.7163 | ... So, to answer your question, **"Love Yourself" is not specifically about loving** |
| | Relevance | Justin Bieber's song 'Love Yourself,' ... widespread media reports and fan speculation suggest it references his past relationship **with singer Rihanna**. | 0.9354 ↑ | Based on the provided context, ... it references Justin Bieber\'s past relationship with singer Rihanna. Answer is: **Rihanna**. |
| | Irrelevance | In the video game Grand Theft Auto: San Andreas, the 'Hot Coffee' mod is ... | 0.6026 ↓ | I cannot provide a response that is not based on the provided context ... |

Figure 3: Example of an LLM's confidence changes when presented with different contexts.

or near-optimal performance on both NQ and HotpotQA datasets. (2) When the downstream LLM was Qwen2.5-7B-Instruct, RAG systems with the fine-tuned or original Reranker showed comparable accuracy, with no significant differences—this partially demonstrates the robustness of bge-reranker-v2-m3-ft.

**Dynamic Retrieval Efficiency.** We evaluated the impact of the dynamic retrieval system CBDR-based on downstream LLM confidence-on RAG performance, alongside related approaches DRA-GIN and CtrlA. Using the NQ_Rerank test set, we measured system accuracy and the fraction of retrieval overhead saved by skipping retrieval under various configurations. Whenever the dynamic module triggered retrieval, documents were directly passed to the reranker, bypassing the full retrieval pipeline. All experiments used the parameter settings recommended by DRAGIN and CtrlA to ensure reproducibility. Results (Table 3) show that: (1) DRAGIN and CtrlA incur low offline cost but higher online inference overhead; (2) their accuracy improves monotonically with retrieval rate; and (3) CBDR reduces retrieval cost by assessing LLM answer confidence-yielding a +0.9 pp accuracy gain under Top-3 reranking versus always retrieving, but a slight -0.2 pp drop in the Top-1 setting.

## 5 DISCUSSION

### 5.1 CONFIDENCE CHANGES CAN SERVE AS A VALID PREFERENCE SIGNAL

This paper tested the confidence changes of LLMs when presented with different contexts. As shown in Figure 3: when Llama3-8B-Instruct answered questions relying solely on its internal parametric knowledge, its confidence was only 0.7163 and the answer was incorrect; after introducing highly relevant correct documents, its confidence increased significantly to 0.9354, and it ultimately reasoned out the correct answer; in contrast, when low-relevance incorrect documents were introduced, its confidence dropped to 0.6026. Additional examples are provided in Appendix D. This result effectively demonstrates the correlation between confidence changes and the true preferences of LLMs: specifically, increased confidence corresponds to "beneficial documents preferred by LLMs", while decreased confidence corresponds to "harmful documents rejected by LLMs". This is fully consistent with the "confidence-based preference definition" proposed in Section 3.2.1, directly verifying the validity of this preference signal. Furthermore, we find that Table 1 also corroborates this conclusion: the experiment reveals a positive correlation between model parameters and reranking performance—performance generally improves as model size and capability increase. For instance, in the Qwen3_Rerank series, though Qwen3_Rerank_8B still has room for improvement in absolute performance, it significantly outperforms its same-series counterpart Qwen3_Rerank_4B and ranks best among all base models. This confirms that the "preference signals" in the NQ_Rerank dataset can effectively distinguish rerankers of different capabilities, and the "stronger capability, better performance" pattern only emerges when the signal logic aligns with the models' true ranking needs—further validating the rationality of using confidence changes as a preference signal.

Thus, the preference-aligned reranker yields substantial performance gains. The fine-tuned `bge-reranker-v2-m3-ft` outperforms all baselines across all metrics, demonstrating that supervised fine-tuning on NQ_Rerank effectively aligns the reranker with the target LLM's (Llama3-8B-Instruct) preferences—specifically, its tendency to select contexts that boost answer confidence. Notably, the gains in Top-1 accuracy ($Precision@1$) and $MRR@1$ underscore its superior ability to retrieve the most critical document with high precision.

## 5.2 EFFECTIVENESS OF PREFERENCE-ALIGNED RERANKER DEPENDS ON TARGET LLM

We find that the optimization effect of the fine-tuned preference-aligned reranker demonstrates significant model dependence, and it can only achieve its full efficacy when paired with the target LLM. As shown in Table 2, pairing bge-reranker-v2-m3-ft with Llama3-8B-Instruct yields a maximum accuracy improvement of 4.7 pp for the RAG system; in contrast, no significant change in accuracy is observed when it is paired with Qwen2.5-7B-Instruct. In essence, this phenomenon stems from the strong anchoring effect of the fine-tuning data on the preferences of the target LLM. In the present study, NQ_Rerank dataset directly constructs labels based on the confidence variations of Llama3-8B-Instruct toward different retrieved contexts. This design enables the reranker to learn, during training, how to filter retrieved contexts that align with the cognitive preferences of the target LLM, with its filtering logic deeply coupled to the target LLM's preferences.

To validate this core logic, a cross-model preference comparison experiment was conducted (see Appendix G for details). A total of 100 uniform sample sets were selected, covering three categories of retrieved contexts: Type A (high semantic relevance but sparse factual details), Type B (comprehensive factual details but marginally lower semantic matching), and Type C (irrelevant contexts). As illustrated in Figure 4, the results reveal substantial differences in the two models' cognitive preferences: Llama3-8B-Instruct favors both Type A and Type B contexts, only strongly rejecting Type C irrelevant contexts; conversely, Qwen2.5-7B-Instruct solely prefers Type A contexts, mildly rejects Type B contexts, and consistently strongly rejects Type C irrelevant contexts. Since the reranker's filtering logic is fully aligned with Llama3-8B-Instruct's preference characteristics, it prioritizes both Type A and Type B contexts—an outcome notably misaligned with Qwen2.5-7B-Instruct's cognition, ultimately rendering the reranker ineffective. This result further corroborates that the preference-aligned reranker's improvement on the RAG system is highly dependent on the consistency between its filtering logic and the target LLM's cognitive preferences.

Notably, although this reranker is trained on the NQ dataset, its performance improvement is not limited in closed book: it also achieves a maximum accuracy improvement of 1.4 pp on the HotpotQA dataset when paired with Llama3-8B-Instruct. This demonstrates the robustness of the preference-aligned reranker: it is truly aligned with the preferences of the target LLM, rather than merely overfitting to the NQ dataset.

## 5.3 DYNAMIC RETRIEVAL BALANCES PERFORMANCE AND EFFICIENCY

By integrating the CBDR, the RAG system can not only maintain competitive accuracy but also significantly reduce retrieval overhead. As shown in Table 3: under the Top-3 setting with $\beta = 0.98$, the accuracy increased by 0.9 pp compared to the baseline, while the retrieval cost decreased by 7.1 pp simultaneously; under the Top-1 setting with $\beta = 0.95$, although the accuracy only decreased slightly by 0.2 pp compared to the baseline, the retrieval cost dropped significantly by 16.7 pp. This result not only fully verifies the significant practical potential of CBDR in balancing efficiency and effectiveness in real-world applications but also confirms the core hypothesis that "the confidence of LLMs can guide retrieval decisions". From the perspective of application value, this balancing feature is crucial for the scenario adaptation of RAG systems by adjusting the value of $\beta$, as show in figure 8 flexible solutions can be provided for different engineering requirements.

## 5.4 LET THE TARGET LLM DECIDE WHETHER TO PERFORM RETRIEVAL

For CBDR, as shown in Table 3, the optimal accuracy under the Top-3 setting is achieved when $\beta = 0.98$: when $\beta$ increases from 0.95 to 0.98 (triggering retrieval for more questions), the accuracy rises by 1.7 pp; however, when $\beta$ reaches 1.00 (forcing retrieval for all questions), the accuracy instead decreases by 0.9 pp. Our phase difference analysis further reveals the reason for this phenomenon (see Appendix F for details): (1) When $\beta$ increases from 0.95 to 0.98: via expanding external knowledge through retrieval, 21 previously incorrect answers were corrected, yet 4 correct answers turned incorrect due to conflicts between external documents and the LLM's inherent parametric knowledge; (2) When $\beta$ increases from 0.98 to 1.00: the above two values become 2 and 11, respectively. Evidently, when $\beta$ changes from 0.98 to 1.00, the benefits from introducing external knowledge are less than the errors caused by knowledge conflicts. This phenomenon directly verifies the hypothesis proposed in Section 3.4: when an LLM has high confidence in answering a question, introducing external knowledge may increase the risk of hallucinations. This conclusion

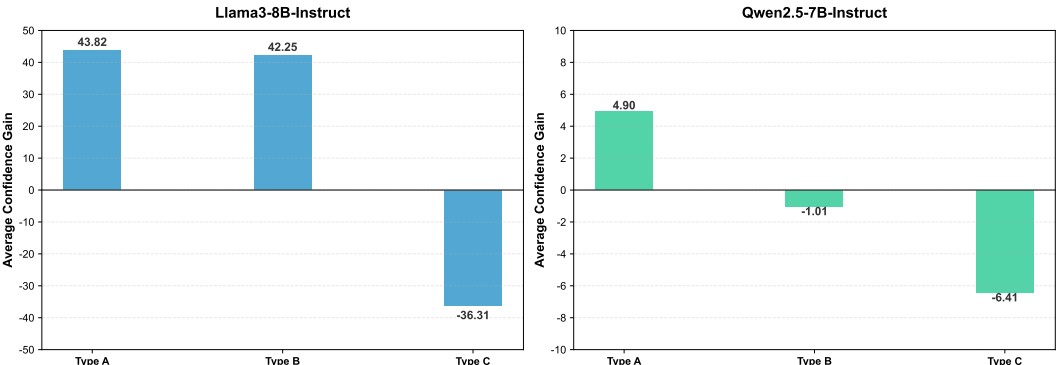

Figure 4: Preference Differences Between Llama3-8B-Instruct (Left) and Qwen2.5-7B-Instruct (Right) for Three Types of Retrieved Contexts.

provides key guidance for the engineering tuning of the $\beta$ value: there is no need to blindly pursue "full retrieval"; instead, a reasonable threshold should be set based on the confidence characteristics of the target LLM to balance the effect of error correction and the risk of hallucinations.

### 5.5 ANALYSIS OF ADDITIONAL RESOURCE OVERHEAD OF THE CBDR FRAMEWORK

This subsection analyzes the additional resource overhead of the CBDR framework, covering the offline and online inference phases. 1) **Offline Phase:** Overhead focuses on two operations. First, dataset construction (NQ_Confidence, NQ_Rerank) relies on the target LLM's forward computation—only requiring hidden vectors before the first token is generated, thus avoiding the high overhead of LLM generation. Second, training/fine-tuning lightweight models: the confidence detection model E (2M-parameter MLP) and reranker (bge-reranker-v2-m3, 568M parameters) have minimal costs, leading to manageable overall computation. Using Llama3-8B-Instruct as the target model, all offline operations can be completed within 6 hours on one NVIDIA RTX 4090 GPU. 2) **Online Inference Phase:** Since the fine-tuned reranker is already part of the RAG system, only one forward pass each by the target LLM and confidence detection model E is needed to determine retrieval necessity—this design ensures online inference fully meets real-time demands.

As shown in Table 3, we compared the additional time consumption of our approach with dynamic retrieval frameworks such as DRAGIN and CtrlA. DRAGIN requires no extra consumption during the offline phase but incurs significant latency during online inference due to real-time uncertainty calculations for generated tokens. CtrlA demands some offline consumption for extracting two features, yet its inference time is slightly less than DRAGIN's because of relatively efficient uncertainty computations. In contrast, although our CBDR introduces higher but manageable offline costs, it significantly reduces online inference time by needing only a single forward pass through both the target LLM and the confidence probe model. In summary, the CBDR framework's additional resource overhead is generally manageable. Notably, the online phase adds negligible latency; further, if retrieval knowledge is parameter-injected (Dong et al. (2025b); Su et al. (2025)), the online forward passes are not additional overhead. For details, see Appendix H.

## 6 CONCLUSION

This paper establishes internal hidden state confidence dynamics as a principled signal for optimizing RAG systems. By quantifying confidence shifts induced by retrieved contexts, we enable precise Reranker alignment and adaptive retrieval activation. Our framework CBDR has brought more efficient performance to the RAG system, which has practical application value. Our work bridges parametric and external knowledge while providing a generalizable paradigm for evaluating knowledge boundary interactions. Future work will extend this approach to multimodal RAG and complex retrieval tasks.

## 7 ETHICS STATEMENT

All authors of this paper adhere to the ICLR Code of Ethics (https://iclr.cc/public/CodeOfEthics). This study uses publicly available datasets, including [Natural Questions (NQ) dataset (Kwiatkowski et al. (2019)) and the HotpotQA dataset (Yang et al. (2018))]. We obtained the datasets in compliance with their official licensing agreements.

## 8 REPRODUCIBILITY STATEMENT

**Source Code:** We have submitted the relevant code used in this paper in the supplementary materials. The code has been detailed to ensure that each key step can be reproduced.

**Dataset Details:** For the NQ_Rerank dataset we proposed, its processing workflow has been described in detail in this paper; furthermore, relevant examples of the dataset can be found in Appendix E.

**Experimental Parameters:** All critical hyperparameters are clearly specified in Appendix C.

**Result Verification:** We have retained the raw data and intermediate results of key evaluations to verify the stability of the results.

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

## A USE OF LLMS

In this paper, Large Language Models (LLMs) were solely employed for the polishing of English writing; they were not involved in any work related to research conceptualization or experiments.

## B PROMPT

In this paper, distinct prompts were utilized to guide LLMs in reasoning for different tasks. In the question-answering scenario, we classify the prompts into two types:

(1) As shown on the left side of Figure 5, this prompt guides the model to directly answer question using its parametric knowledge. When constructing the dataset for the Confidence Detection Model E, this prompt was consistently used to guide the Llama3-8B-Instruct model in generating answers. Additionally, this prompt is applied in scenarios within the CBDR system where the LLM has high confidence in answering this question and thus skips the retrieval step to provide a direct response.

(2) As shown on the right side of Figure 5, this prompt guides the LLM to answer question by combining external knowledge with its parametric knowledge. When constructing the preference dataset NQ_Rerank, this prompt was used to guide Llama3-8B-Instruct to generate the first Token based on the provided context; meanwhile, this prompt is also utilized in the CBDR system when the LLM needs to answer question with reference to retrieved documents.

## C IMPLEMENTATION DETAILS.

During the training of the Confidence detection model E, the initial learning rate was set to $5e^{-5}$, the dropout rate was configured to 0.5, and the training was conducted over 30 epochs. For the fine-tuning of the bge-reranker-v2-m3 model, the initial learning rate was set to $6e^{-5}$, weight decay was configured to 0.01, the maximum query length (query_max_len) was set to 128, the maximum passage length (passage_max_len) was set to 512, and the training was performed for 1 epoch.

| QA Prompt | RAG Prompt |
|---|---|
| You need to read the question carefully and answer it based on your own knowledge.

Question: {question} | You are a rigorous language model. Please answer the question based on the provided context. If the context does not support reasoning about the answer, please answer the question based on your own knowledge.

Contexts: {contexts}
Question: {question} |

Figure 5: The prompt on the left side of the figure guides the LLM to answer question using its parametric knowledge; the prompt on the right side requires the LLM to answer question by combining external knowledge with its parametric knowledge.

## D   LLM CONFIDENCE

The training of the Confidence Detection Model E fully follows the approach described in (Ni et al. (2025)), and the downstream LLM employed is Llama3-8B-Instruct. After obtaining Model E, we conducted an initial verification to examine how the LLM's confidence in answering questions varies when provided with relevant versus irrelevant documents. As illustrated in Figure 6, the observed changes in confidence confirm our hypothesis: changes in the LLM's internal hidden states can guide the selection of external knowledge. It should be noted that in this paper, external knowledge refers specifically to contexts obtained through retrieval.

## E   PREFERENCE DATASET

In this paper, to align the preferences of the Reranker with the target LLM, we constructed the preference dataset NQ_Rerank using changes in the LLM's confidence. The NQ_Rerank dataset is divided into a training set with 7,622 items and an evaluation set with 1,216 items. As shown in Figure 7, each data item contains four fields: query, pos, neg, and prompt. Among these, pos and neg are lists of positive and negative contexts, respectively. Specifically, each context in pos enhances the LLM's confidence, whereas each context in neg reduces the LLM's confidence. The prompt field refers to the default prompt used by the Reranker model for the reranking task.

## F   DYNAMIC RETRIEVAL

When CBDR is adopted for dynamic retrieval in the RAG system (The algorithm can be found in Algorithm 1), the system achieves the optimal performance at $\beta = 0.98$. As illustrated in Figure 8, this figure comprehensively depicts the impact of the threshold $\beta$ (ranging from 0.90 to 1.00) on both RAG system performance (quantified by the system score on the left y-axis) and retrieval efficiency (measured by the skip retrieval ratio on the right y-axis). To further elucidate the performance characteristics around this optimal threshold, we analyzed the differential responses corresponding to two threshold intervals: $\beta$ ranging from 0.95 to 0.98 and from 0.98 to 1.00. Specifically, these differential responses refer to the questions where the integration of external knowledge (enabled by dynamic retrieval) alters the correctness of the LLM's final answers.

Our observations of these data reveal the following: When the introduction of external knowledge enables the LLM to change from answering incorrectly to correctly, this is always because the introduced external knowledge expands the knowledge boundary of the RAG system, thereby leading the LLM to generate correct answers. However, when the introduction of external knowledge instead causes the LLM to shift from answering correctly to incorrectly, there are multiple reasons for this: (1) As shown in Figure 9, the most common reason is the introduction of incorrect external knowledge, which causes conflicts between the model's internal parametric knowledge and the external knowledge, thereby triggering hallucinations. (2) As shown in Figure 10, the introduced

| Question | Context_type | Context | Confidence |
|---|---|---|---|
| when was the last time anyone was on the moon | No | | 0.7454 |
| | Relevance | The Apollo program by NASA included the last human Moon landing during Apollo 17 in December 1972. Astronauts Eugene Cernan and Harrison Schmitt landed on the lunar surface on 14 December 1972 UTC, conducting three days of exploration. No subsequent human missions have reached the Moon since then, making this the final | 0.8562 ↑ |
| | Irrelevance | In the video game Grand Theft Auto: San Andreas, the 'Hot Coffee' mod is an unauthorized user modification that accesses a hidden mini-game featuring sexual interactions. This content was originally inaccessible in the official release but was discovered in the game's code, leading to controversy and a re-rating of the game by the ESRB. | 0.6293 ↓ |
| when did the eagles win last super bowl | No | | 0.9288 |
| | Relevance | The Philadelphia Eagles of the NFL last won the Super Bowl in February 2018, which corresponded to the 2017 league season. They defeated the New England Patriots 41-33 in Super Bowl LII, securing their first championship since 1960. As of the current time in 2025, this remains their most recent Super Bowl victory. | 0.9462 ↑ |
| | Irrelevance | In the video game Grand Theft Auto: San Andreas, the 'Hot Coffee' mod is an unauthorized user modification that accesses a hidden mini-game featuring sexual interactions. This content was originally inaccessible in the official release but was discovered in the game's code, leading to controversy and a re-rating of the game by the ESRB. | 0.4606 ↓ |
| how many seasons of the bastard executioner are there | No | | 0.5645 |
| | Relevance | FX's historical drama series 'The Bastard Executioner,' created by Kurt Sutter, premiered in September 2015 but received low ratings and mixed reviews. Consequently, the network canceled it after the first season's conclusion in November 2015, with no renewal for additional seasons. | 0.9410 ↑ |
| | Irrelevance | Justin Bieber's song 'Love Yourself,' released in 2015, features lyrics co-written with Ed Sheeran that are interpreted as addressing an ex-partner. Although unconfirmed directly by Bieber, widespread media reports and fan speculation suggest it references his past relationship with singer Rihanna, contributing to the song's narrative. | 0.5078 ↓ |

Figure 6: This figure presents three examples of confidence changes. For each example, the confidence levels indicated by the LLM's internal hidden states are provided under three scenarios, namely: without external documents provided, with relevant documents provided, and with irrelevant documents provided.

external knowledge contains correct documents, but the model exhibits attention bias and fails to focus on these correct documents. (3) As shown in Figure 11, the questions are time-sensitive. This issue arises due to the inherent temporal limitations of the NQ dataset; thus, the NQ dataset provides outdated external documents and reference answers. As a result, the LLM would originally answer correctly, but ends up answering incorrectly due to the erroneous external knowledge.

# G    CROSS-MODEL PREFERENCE COMPARISON EXPERIMENT

This experiment aims to quantify the preference differences between Llama3-8B-Instruct (the target model) and Qwen2.5-7B-Instruct (the comparison model) toward different types of retrieved contexts. It verifies the strong binding characteristic between the filtering logic of the preference-aligned reranker and the target LLM at the cognitive mechanism level, providing empirical support for the core conclusion in the main text that the reranker's effectiveness is model-dependent.

## G.1    TARGET LLMS

1) **Llama3-8B-Instruct:** Fine-tuning target of the preference-aligned reranker and baseline model in the main context experiments; 2) **Qwen2.5-7B-Instruct:** An open-source LLM of the same scale, used to verify the generality of preference differences.

| Query | who does sam neil play in peter rabbit |
|---|---|
| Pos | ["Peter Rabbit is a 2018 live-action/computer-animated comedy film directed by Will Gluck and written by Rob Lieber and Gluck, based on the stories of Peter Rabbit created by Beatrix Potter. ...",

"... The accusations focused on a scene where Thomas McGregor \u2014 whose character has a known severe allergy to blackberries \u2014 is pelted with the berries until one enters his mouth, causing him to enter anaphylactic shock and grab for his Epipen.[35][36][37] ...",

"... A local toy shop on Compston Road, Ambleside, was adapted to be Mr McGregor's.[citation needed]"] |
| Neg | ["The film was first revealed in April 2015 through email leaks as a result of the Sony Pictures hack.[8] The official announcement of the film came that December.[9]",

"The Singing Sparrows were voiced by Jessica Freedman, Shana Halligan, Katharine Hoye, Chris Mann, Chad Reisser, and Fletcher Sheridan",

"Peter feels bad for what he has done, and upon learning that Bea intends to leave the neighborhood, he and Benjamin head to London to find Thomas at Harrods...",

"Thomas and Peter start a war with each other by setting up traps and other offensive nuisance..."] |
| Prompt | Given a question, retrieve Wikipedia passages that answer the question. |

Figure 7: This figure presents an example of the preference dataset NQ_Rerank; each data item contains four fields: Query, Pos, Neg, and Prompt.

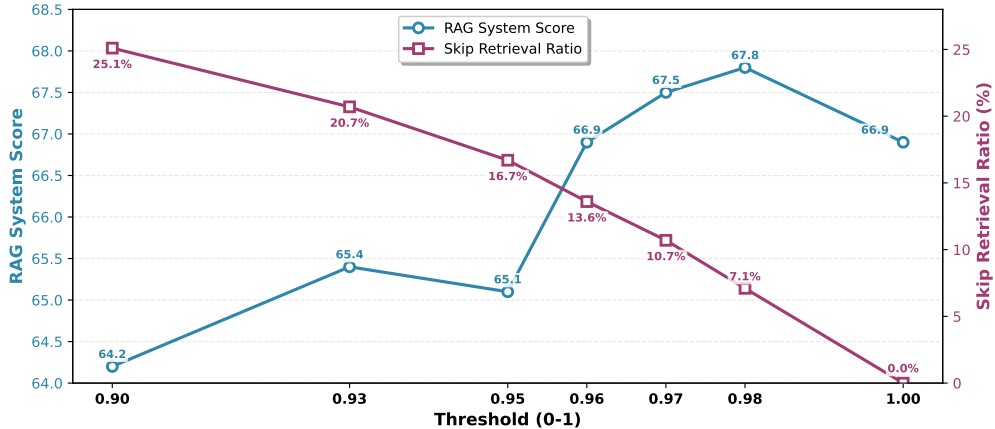

Figure 8: The Impact of Threshold (0–1) on RAG System Performance and Retrieval Efficiency under the Top-3 Setting. X-axis: Threshold (0.90–1.00); Left Y-axis: RAG System Score; Right Y-axis: Skip Retrieval Ratio (%).

## G.2 RETRIEVED CONTEXT CORPUS

The corpus is randomly selected and constructed from candidate retrieved contexts in the NQ dataset, consisting of 100 sample sets. Each set corresponds to one unified question and includes 3 types of retrieved contexts (1 context per type, generated by rewriting gold contexts), ensuring a single controllable experimental variable: 1) **Type A:** Highly matches the core semantics of the question but only mentions core concepts without specific data, logical chains, or supplementary details; 2) **Type B:** Contains complete factual information (data, logic, etc.) required to answer the question but has slightly lower semantic matching with the question's expression (consistent core concepts, but expressed through synonymous substitution and sentence structure reconstruction); 3) **Type C:** Extremely low semantic relevance to the question and no question-related factual information, serving only as a control benchmark for preference judgment.

| Question | Context | Retrieval | Reference | Answer |
|---|---|---|---|---|
| when did brazil last host the fifa world cup | ["The **1951** World Cup hosting duty was decided on 26 July **1946**, the same day that Brazil was selected for the **1949** World Cup, in Luxembourg City. On 27 July ...", | No | | According to my knowledge, Brazil last hosted the FIFA World Cup in **2014**. |
| | "Brazil made the first unopposed bid since the initial selection of the **1986** FIFA World Cup (when Colombia was selected as host, but later withdrew for financial problems) ... "Only Mexico, Italy, France, Germany (West Germany until shortly after the **1990** World Cup) and Brazil have hosted the event on two occasions ..."] | Top-3 | **2014** | Based on the provided context, it can be inferred that Brazil last hosted the FIFA World Cup in **1950**. |

Figure 9: Since all the introduced external knowledge is incorrect, it causes the LLM's answer to shift from correct to incorrect.

| Question | Context | Retrieval | Reference | Answer |
|---|---|---|---|---|
| when was the art institute of chicago founded | ["The Art Institute of Chicago, founded in **1879** and located in Chicago's Grant Park, is one of the oldest and largest art museums in the United States ...", | No | | The Art Institute of Chicago was founded in **1879** ... |
| | "In **1866**, a group of 35 artists founded the Chicago Academy of Design in a studio on Dearborn Street, with the intent to run a free school with its own art gallery ... The Academy's charter was granted in March **1867**.", "In **1882**, the Chicago Academy of Fine Arts changed its name to the current Art Institute of Chicago and elected as its first president the banker and philanthropist Charles L ..."] | Top-3 | **1879** | Based on the provided context, the Art Institute of Chicago was founded in **1866** as the Chicago Academy of Design ... |

Figure 10: Although the external knowledge contains correct context, the LLM focuses on incorrect contexts, resulting in its answer shifting from correct to incorrect.

| Question | Context | Retrieval | Reference | Answer |
|---|---|---|---|---|
| who was the most recent president to be impeached | ["... Andrew Johnson and **Bill Clinton** are the only two presidents to have been successfully impeached ...", | No | | The most recent president to be impeached was **Donald Trump**, the 45th President of the United States ... |
| | "... President **Richard Nixon** resigned from office after the House Judiciary Committee had already reported articles of impeachment to the floor.", "... especially of presidents, are common,[22][23] going back to the administration of **George Washington** in the mid-1790s ..."] | Top-3 | **Bill Clinton** | ... the most recent president to be impeached was **Bill Clinton**, who was successfully impeached by the House of Representatives in 1998 ... |

Figure 11: Since the questions are time-sensitive, the contexts and reference provided by the NQ dataset are all incorrect; the retrieved external knowledge thus causes the LLM to answer incorrectly.

### G.3 CONTROLLED VARIABLES

To avoid interference from irrelevant factors, the following variables are strictly controlled for all contexts: 1) **Context Length:** Uniformly limited to within 200 characters; 2) **Format Standardization:** Pure context format without special symbols, lists, formulas, etc.; 3) **Language Style:** Formal written language, avoiding colloquial or emotional expressions; 4) **Question Consistency:** The 3 context types in each sample set correspond to the same question, ensuring that preference differences arise only from context types rather than the question itself.

---

**Type A Prompt**

Based on the following Query and corresponding key factual text (Gold content), generate an English auxiliary response text.Requirements: 1) Fully retain high semantic relevance to the query without deviating from the core topic; 2) Only keep core concepts and conclusions, and delete all specific data, logical reasoning processes, and case details; 3) Control the length to approximately 1/3 of the Gold content with concise language; 4) Return the result in JSON format.

Example:
Query: "What is the approximate length of the Amazon River in kilometers?"
Content: "The Amazon River is the second longest river in the world, with a total length of its main stream about 6,400 kilometers, a drainage basin area of 7.05 million square kilometers, and flowing through 9 countries including Brazil and Peru."
Output:
```

json

{
    "result": "The Amazon River is one of the world's major long rivers, with its main stream totaling approximately several thousand kilometers and flowing through multiple South American countries."
}
```

Input:
Query: {question}
Gold content: {content}
Output:

---

Figure 12: Type A prompt is used to generate content that is semantically relevant to the gold context but with concise factual details.

### G.4 EXPERIMENT

This experiment adopts a baseline-test control design to quantify model preferences through confidence changes. The specific steps are as follows:

**Sample Preprocessing:** Standardize the 100 sets of questions and their corresponding Type A, B, and C contexts: Use Prompts to guide LLMs in generating contexts based on the questions and Gold context; the Type A prompts are shown in Figure 12.

**Confidence Collection:** Adopt a dual-scenario comparison of "context-free baseline" and "context-augmented test" to collect the generation confidence of the two models: Context-Free Baseline (Base_Conf): Input the 100 standardized questions individually into Llama3-8B-Instruct and Qwen2.5-7B-Instruct. The models generate answers relying solely on internal parameter knowledge, and the confidence during generation is recorded (calculated as the maximum value of the softmax outputs of model logits, with a value range of [0,1]). Context-Augmented Test (Test_Conf): For each sample set, construct three input formats—"question + Type A context", "question + Type B context", and "question + Type C context"—and input them into the two LLMs sequentially. After generating answers, the corresponding confidence is recorded (using the same calculation method as the baseline).

**Data Calculation and Statistics:** Conduct quantitative analysis on the collected confidence data, with the core calculation logic as follows: Per-Sample Confidence Improvement Value: For each model, sample set, and text type, calculate Improvement Value = Test_Conf - Base_Conf. Positive value indicates the context enhances model confidence, i.e., "preference"; Negative value indicates the context reduces model confidence, i.e., "rejection". Average Confidence Improvement Value: For each model and context type, compute the total sum of improvement values across the 100 sample sets.

# H    DETAILED ANALYSIS OF RESOURCE CONSUMPTION FOR THE CBDR FRAMEWORK

The additional resource consumption required to construct the CBDR framework is divided into the offline preparation phase and online inference phase, with overall computational costs well-controlled. Details are as follows:

## H.1    OFFLINE PREPARATION PHASE

The offline phase involves four core tasks, with resource consumption concentrated on target LLM forward computations and lightweight model training (no generation operations), ensuring manageable computational costs: 1) **Constructing the NQ_Confidence dataset:** Provides training data for the confidence detection model E, including input contexts and corresponding target LLM confidence labels. Sample scale: 2,000 training samples, 1,000 development (Dev) samples, and 500 test samples, totaling 3,500 samples. Computational load: Each sample requires one forward pass of the target LLM (e.g., Llama3-8B-Instruct) to extract confidence features, resulting in a total of 3,500 target LLM forward passes. 2) **Training the confidence detection model E:** Model architecture: 5-layer MLP with 2M parameters. Training configuration: Trained on the NQ_Confidence training set for 100 epochs with a batch size of 32. Resource characteristics: Lightweight model with extremely low training overhead; completable within 10 minutes on a single consumer-grade GPU (e.g., NVIDIA RTX 4090). 3) **Constructing the NQ_Rerank fine-tuning dataset:** Provides fine-tuning data for the preference-aligned reranker, with labels generated based on confidence changes of the target LLM toward different retrieved contexts. Sample scale: 7,622 training samples and 1,216 test samples, totaling 8,838 samples. Computational load: Each sample requires 8 forward passes of the target LLM (matching 8 types of retrieved contexts) to generate preference labels, resulting in a total of 70,704 target LLM forward passes. 4) **Fine-tuning the reranker:** Base model: bge-reranker-v2-m3 (568M parameters). Fine-tuning configuration: Fine-tuned on the NQ_Rerank training set for 5 epochs (weights after 1 epoch are used in practice) with a batch size of 32. Resource characteristics: Moderate-scale model with few fine-tuning epochs; the entire process can be completed within 2 hours on a single NVIDIA RTX 4090.

## H.2    ONLINE INFERENCE PHASE

The additional resource consumption in the online phase only involves two lightweight forward passes, which do not affect inference efficiency: 1) **Workflow:** For input query, the target LLM first performs one forward pass to obtain hidden states, while the confidence detection model E (2M parameters) executes one forward pass to determine the necessity of retrieval. If retrieval is deemed unnecessary, the generation process can be directly connected using the results of this target LLM forward pass, eliminating the need for additional LLM calls and redundant computations. 2) **Optimization:** Retrieval knowledge is directly injected into the target LLM's Attention module (Dong et al. (2025b)) or FNN module (Su et al. (2025)) via parameter injection. Thus, even if retrieval is required after the LLM's forward pass, the generation process can be seamlessly connected following parameterized knowledge injection, with no redundant computational overhead. 3) **Latency:** The total latency of the two forward passes is less than 100ms (on a single NVIDIA RTX 4090), which negligibly increases inference latency and meets real-time requirements.

---

**Algorithm 1:** CBDR Inference

---

**Input:** Query $q$

**Output:** Final answer $y$

1 **Require:** Target LLM $M$, confidence detection model $E$, retriever $\mathcal{R}$, document corpus $\mathcal{D}$, Fine-tuning reranker $FR$, confidence threshold $\beta$, $Top - K$ parameter $K$, QA prompt $\mathcal{P}_{\text{qa}}$, RAG prompt $\mathcal{P}_{\text{rag}}$;

2 Construct pure QA prompt: $\texttt{prompt}_{\text{qa}} \leftarrow \mathcal{P}_{\text{qa}}(q)$;

3 Tokenize input: $\mathbf{x} \leftarrow \texttt{Tokenize}(\texttt{prompt}_{\text{qa}})$;

4 Forward through $M$ with hidden states output:
  $\texttt{outputs} \leftarrow M(\mathbf{x}, \texttt{output\_hidden\_states} = \texttt{True})$;

5 Extract last context hidden state: $\mathbf{V} \leftarrow \texttt{outputs.hidden\_states}[-1][0, -1, :]$;

6 Compute confidence score: $S \leftarrow E(\mathbf{V})$;

7 **if** $S \geq \beta$ **then**

8  Generate answer directly: $y \leftarrow M.\texttt{generate}(\mathbf{x}, \texttt{max\_new\_tokens} = L_{\max})$;

9 **else**

10  Retrieve candidate documents: $\mathcal{D}_{\text{raw}} \leftarrow \mathcal{R}.retrieve(q, \texttt{top\_k} = K \cdot r)$;

11  Rerank using $M$-aligned reranker: $\mathcal{D}_{\text{reranked}} \leftarrow FR.\texttt{rerank}(q, \mathcal{D}_{\text{raw}}, K)$;

12  Construct context: $\texttt{context} \leftarrow \bigoplus_{i=1}^{K} \mathcal{D}_{\text{reranked}}[i].\texttt{text}$;

13  Build RAG prompt: $\texttt{prompt}_{\text{rag}} \leftarrow \mathcal{P}_{\text{rag}}(\texttt{context}, q)$;

14  Tokenize RAG input: $\mathbf{x}_{\text{rag}} \leftarrow \texttt{Tokenize}(\texttt{prompt}_{\text{rag}})$;

15  Generate answer with evidence: $y \leftarrow M.\texttt{generate}(\mathbf{x}_{\text{rag}}, \texttt{max\_new\_tokens} = L_{\max})$;

16 **end**

17 Return $y$;

---

