# OpenReview forum: "Rethinking LLM Parametric Knowledge as Confidence for Effective and Efficient RAG"
_ICLR.cc/2026/Conference — ICLR 2026 Conference Withdrawn Submission_

### Official Review · Reviewer_T2av · 2025-10-25

**Soundness:** 3
**Presentation:** 2
**Contribution:** 2
**Rating:** 4
**Confidence:** 5

**Summary:**

This paper investigates how to effectively quantify and exploit the parametric knowledge boundaries of LLMs for improving RAG effectiveness and efficiency. The authors propose a confidence-shift metric derived from LLM hidden states to rerank retrieved contexts, fine-tune a reranker on the synthetic NQ_Rerank dataset, and introduce CBDR to gate retrieval adaptively. The confidence-shift formulation is clearly presented.  Experiments on NQ and HotpotQA validate the anticipated behavior, showing accuracy improvements with reduced retrieval cost.

**Strengths:**

1. Clarity of presentation: The overall narrative is coherent, with well-motivated objectives and a logically developed framework.
2. Clear methodological formulation: The confidence-shift framework is precisely defined that facilitates understanding.
3. Comprehensive discussion and analysis: The paper provides thorough discussions with detailed analyses of confidence variation, offering valuable insights into how confidence signals guide retrieval.

**Weaknesses:**

1. Limited coverage of related work: The discussion of related work is insufficient. For example, while Section 2.1 discusses the issue of neglecting potential conflicts between external and internal knowledge, the citation coverage is incomplete—it overlooks several closely related studies (e.g., Parenting[1] ).
2. Limited experimental validation:
	•	The paper would benefit from more extensive experiments. In particular, it lacks evaluation in different hyperparameter settings, and no ablation studies of individual components are provided.
	•	Limited performance gains in some settings: The method shows modest improvements in certain cases;  for example, in Table 3, Qwen2.5-7B-Instruct on NQ Top-1 and Llama3-8B-Instruct on HotpotQA Top-1 do not outperform other methods.
	•	Minor issues in presentation and details: Some presentation details need improvement. For example, there are typos (e.g., “a taget LLM” in line 441), and the notation for highlighting results is not explained (e.g., bold for best, underline for second best).

[1] Xu Y, Zhang R, Jiang X, et al. Parenting: Optimizing Knowledge Selection of Retrieval-Augmented Language Models with Parameter Decoupling and Tailored Tuning[J]. CoRR, 2024.

**Questions:**

1. Providing pseudocode: Including pseudocode could help readers better understand the proposed method and clarify its implementation details.
2. Comparison with existing adaptive retrieval methods: Since the proposed method features adaptive retrieval, it would be helpful to compare its performance with existing adaptive retrieval approaches (e.g., Adaptive RAG[3], Probing RAG[4]) to better evaluate its adaptive retrieval capability.
3. Additional end-to-end comparisons: It would be valuable to compare the proposed method with approaches mentioned in the related work, such as DPA-RAG and SEAKR, under the end-to-end experimental setting.

[3] Jeong S, Baek J, Cho S, et al. Adaptive-RAG: Learning to Adapt Retrieval-Augmented Large Language Models through Question Complexity[C]//Proceedings of the 2024 Conference of the North American Chapter of the Association for Computational Linguistics: Human Language Technologies (Volume 1: Long Papers). 2024: 7029-7043.

[4] Baek I, Chang H, Kim B J, et al. Probing-RAG: Self-Probing to Guide Language Models in Selective Document Retrieval[C]//Findings of the Association for Computational Linguistics: NAACL 2025. 2025: 3287-3304.

---

> ### Author Response · Authors · 2025-11-27
>
> We appreciate your valuable comments, which have not only helped improve our research but also provided important guidance for subsequent related work. Below, we address each of your questions one by one; please feel free to point out any further issues:
>
> 1. Your guidance on related work is highly insightful. We have incorporated additional relevant literature into the revised manuscript, including the references you recommended.
> 2. We have added cross-model preference comparison experiments in the revised manuscript (see details in Lines 444–455) to support our claim that “replacing the target LLM after reranker fine-tuning may lead to performance degradation.” The core reason lies in the strong coupling between the reranker and the target LLM: the fine-tuning labels are constructed directly from the target LLM’s confidence shifts in response to different retrieved contexts. Consequently, a reranker fine-tuned on Llama3-8B-Instruct does not fully align with Qwen2.5-7B-Instruct’s confidence preferences. Although this reranker does not achieve state-of-the-art performance across all settings when paired with Llama3-8B-Instruct, it still yields a significant improvement over the baseline bge-reranker-v2-m3. Additionally, we have included an analysis of the impact of the threshold hyperparameter β on the CBDR system in Appendix F.
> 3. Thank you for your suggestions regarding writing clarity. We have refined the descriptions of Tables 2 and 3 to more clearly convey their key takeaways and have thoroughly proofread the manuscript to correct spelling and grammatical errors, thereby enhancing overall readability.
> 4. Your suggestion regarding pseudocode is very valuable. We have added detailed pseudocode in the revised manuscript (see details in Lines 1046–1064) to explicitly illustrate the core mechanism of CBDR.
> 5. Your recommendation to include end-to-end experiments with full retrieval pipelines is very helpful. In the revised manuscript, we have expanded our evaluation accordingly (results in Table 3). Specifically, we compare DRAGIN, CtrlA, and our CBDR in terms of: (i) the proportion of queries requiring retrieval under varying confidence thresholds, (ii) final RAG accuracy, and (iii) resource consumption during both offline preparation and online inference. While CBDR incurs higher offline costs than DRAGIN and CtrlA due to confidence modeling, its online inference overhead is negligible. Detailed results and discussion can be found in the main text (see details in Lines 509–529).

---

> ### Comment · Reviewer_T2av · 2025-11-27
> **Thanks**
>
> Thanks for your response — it has resolved my concerns.
>
> Noticing the replies from other reviewers, I would suggest the authors include some additional related works in the Related Work section, and update your PDF with the modified colors:
>
> [1] Parenting: Optimizing Knowledge Selection of Retrieval-Augmented Language Models with Parameter Decoupling and Tailored Tuning. ACL 2025. This work also focuses on internal and external knowledge, similar to yours.
>
> [2] TC–RAG: Turing–Complete RAG’s Case Study on Medical LLM Systems. ACL 2025. Which follows internal knowledge to suggest RAG.
>
> I have raised my score from 4 to 6.

---

### Official Review · Reviewer_DHB1 · 2025-10-31

**Soundness:** 2
**Presentation:** 3
**Contribution:** 2
**Rating:** 4
**Confidence:** 4

**Summary:**

This paper proposes a confidence evaluation based approach to improve RAG systems. The method leverages LLM internal hidden states to: (1) construct a confidence detection model that quantifies how retrieved contexts enhance model confidence, (2) build a preference dataset (NQ_Rerank) based on confidence shifts to fine-tune rerankers, and (3) introduce Confidence-Based Dynamic Retrieval (CBDR) to adaptively trigger retrieval based on initial confidence. Experiments on Natural Questions and HotpotQA datasets show improvements in reranking accuracy (5.19%), end-to-end RAG accuracy (4.70%), and retrieval efficiency (7.10% cost reduction with 5.60% accuracy gain).

**Strengths:**

The introduced Confidence Shift metric is intuitive and understandable.

Experimental setup, despite several issues that I will describe in the next section, is well-defined: multiple datasets and metrics are considered. Several reranker models are also considered.

**Weaknesses:**

First and foremost, the overall novelty of the paper is modest and limited. It is obvious that the work heavily relies on prior work by Ni et al. (2025) and it is explicitly stated that “paper’s core innovation is a novel preference metric”. This, while being an important novel contribution, stands as an incremental novelty.

It remains unclear, why do the authors consider only two not state-of-the-art LLMs at-the-time: LLama 3 8B and Qwen 2.5 7B. At the time of the submission, a newer and more capable family of Qwen 3 models were actually available. It would be beneficial to see these models in the submission.

Moreover, the scope of the LLMs considered is narrow with respect to model size: while it is challenging to evaluate 70B models and modern MoE models (Qwen 235B, GLM 4.5-4.6), there are several smaller (Llama 3.2 1B) and only slightly bigger models (Gemini 3 12B). Next, there are several hybrid architectures available that could also be tested (e.g phi-4-mini-instruct).

The proposed method itself introduces a noticeable computational overhead, which is not studied and discussed in the submission. It seems like the proposed method, compared to the standard RAG pipeline, introduces a multiplicative computational overhead: while for original RAG it takes only 1 forward pass to process the input query, the proposed method requires N forward passes where N is the number of retrieved contexts.

**Questions:**

In 158-159 you state that layer 2 is picked for hidden state extraction. Why exactly layer 2? Did you experiment with other layers?

There is no discussion and comparison of the computational overhead of your method. While it is clear from the description that such overhead is present.

In 4.1 in the Datasets subsection you describe the training set for the confidence model E: 1k samples per negative and positive examples, respectively. Have you experimented with different set sizes? What is the minimum dataset size required for convergence? Is 50/50 negative/positive partition of the training data necessary?

In 5.2 you state that the fine-tuned reranker “shows no statistical significance” when used with Qwen 2.5 7B - does it mean that the reranker is highly dependent on the target model? If so, is there a way to make the reranker model-agnostic? If not, I strongly suggest adding the aforementioned weakness as a limitation and include this statement in the paper.

In Table 2 `bge-reranker-v2-m3 ` outperforms yours fine-tuned `bge-reranker-v2-m3-ft` on the NQ dataset, which is counter-intuitive. Can you elaborate on that?

In Table 3 you test only three different $\beta$ threshold values. Why only these? Can you experiment with lower $\beta$ values? What is the minimum threshold?

---

> ### Author Response · Authors · 2025-11-21
>
> We appreciate your valuable comments, which have not only helped improve our research but also provided important guidance for subsequent related work. Regarding the innovation of the research, although "evaluating confidence using model hidden vectors" has been explored in existing studies, to the best of our knowledge, this is the first study to use "confidence shifts of the model when exposed to different retrieved documents" as a preference signal and apply it to RAG systems to enhance performance. Below, we address your remaining questions one by one; please feel free to point out any further issues:
>
> 1. In this study, we select the hidden vectors from the Layer/2 layer of the target LLM before generating the first token as the carrier for confidence extraction. Taking the Llama3-8B-Instruct model (with 32 hidden layers) as an example, the 16th-layer hidden vectors are specifically extracted. This selection is based on the verification results of existing work, as detailed in Section 3.1.1 of the manuscript.
>
> 2. Your suggestion on computational overhead is highly valuable. We have discussed the relevant overhead in detail in Lines 512-524 of the revised manuscript, which is divided into two parts: 1) Offline Preparation Phase: Constructing the training dataset for the confidence detection model E and the reranker fine-tuning dataset requires only approximately 74,204 forward passes of the target LLM. The two models have 2M and 568M parameters, respectively, resulting in low training and fine-tuning overhead. For the Llama3-8B-Instruct model, the full offline preparation work (including data preparation, training and fine-tuning, and reading/writing) can be completed within 6 hours on a single NVIDIA RTX 4090 GPU. 2) Online Inference Phase: The fine-tuned reranker is already part of the RAG system. Only one forward pass each by the target LLM and the confidence detection model E is required to determine the necessity of retrieval. For the Llama3-8B-Instruct model, the additional latency introduced by this process does not exceed 100ms.
>
> 3. The 50/50 split ratio of the training set is primarily intended to alleviate the adverse impact of label imbalance on model convergence. Currently, we have not tested other dataset sizes.
>
> 4. As you pointed out, the fine-tuned reranker exhibits strong dependence on the target model—the labels of the reranker’s fine-tuning dataset are directly constructed based on the target LLM’s confidence changes toward different retrieved contexts. We have detailed this correlation in the revised manuscript and supplemented a cross-model preference comparison experiment to corroborate it (see details in Lines 447-458); we have also clearly stated this research limitation in the Discussion section.
>
> 5. The phenomenon in Table 2 where "bge-reranker-v2-m3 outperforms bge-reranker-v2-m3-ft" only occurs when the downstream model is Qwen2.5-7B-Instruct. We have explored the reason in the revised manuscript: bge-reranker-v2-m3-ft is fine-tuned based on the confidence preference signals of the Llama3-8B-Instruct model. When the downstream model is replaced with Qwen2.5-7B-Instruct, the mismatch in model preferences leads to a decrease in RAG system accuracy.
>
> 6. We have tested various β thresholds. Due to space constraints in the table, only 3 representative thresholds are presented in Table 3. We have detailed the influence of threshold changes (0.90 - 1.00) on RAG system performance in Appendix F of the revised manuscript.

---

> > ### Comment · Reviewer_DHB1 · 2025-11-26
> > **Reviewers Answer**
> >
> > I am grateful to the authors for the detailed and extensive response to my review. While I appreciate the given clarifications on some of the weaknesses pointed out, there are also problems with this submission that remain unaddressed. I will only quote the parts of your response that seem unaddressed for me. If some part of your rebuttal is not quoted that means I am satisfied with it.
> >
> > > "evaluating confidence using model hidden vectors" has been explored in existing studies, to the best of our knowledge, this is the first study to use "confidence shifts of the model when exposed to different retrieved documents" as a preference signal and apply it to RAG systems to enhance performance
> >
> > Yes, you can claim this approach is novel. However, I still strongly believe it represents an incremental extension of previous confidence detection method [1] rather than something ideologically novel. The core idea of extracting confidence from hidden states is not firstly introduced in this work.
> > > The 50/50 split ratio of the training set is primarily intended to alleviate the adverse impact of label imbalance on model convergence. Currently, we have not tested other dataset sizes.
> > I was hoping to see the test of other splits as a part of the ablation. 2k total samples is a small training set for a neural classifier. Since no ablation (or a solid justification of its absence) is presented, this weakness is still not addressed.
> > > As you pointed out, the fine-tuned reranker exhibits strong dependence on the target model—the labels of the reranker's fine-tuning dataset are directly constructed based on the target LLM's confidence changes toward different retrieved contexts. We have detailed this correlation in the revised manuscript and supplemented a cross-model preference comparison experiment to corroborate it (see details in Lines 447-458); we have also clearly stated this research limitation in the Discussion section
> > I appreciate the transparency about this limitation. However, this model dependency severely limits practical applicability. So this weakness remains as a limitation of this work.
> > —
> >
> > Next, there is still an absence of statistical significance in your response, despite being a serious drawback. Without significance tests, the readers can not clearly see if the reported improvements are statistically meaningful or are within noise margins. This is a crucial requirement for a solid research paper.
> >
> > Furthermore, there is an absence of the exploration of other LLM sizes and different types of architectures (e.g. SSMs, Hybrid models). So the concerns about generalizability of the method remain.  Will the method work for larger models? Different architectures?
> > —
> > All that being said, while I am grateful for the extensive rebuttal, there are still major concerns that remain unaddressed for this submission. Therefore, I prefer to keep my evaluation score as is.

---

### Official Review · Reviewer_emZB · 2025-11-02

**Soundness:** 2
**Presentation:** 3
**Contribution:** 2
**Rating:** 2
**Confidence:** 4

**Summary:**

This paper addresses the challenge of knowledge boundary awareness in Retrieval-Augmented Generation (RAG) by introducing a novel knowledge probing approach that leverages the continuous hidden states of Large Language Models (LLMs) to enhance both effectiveness and efficiency. The key technical contributions are twofold: first, a confidence detection model that quantifies how retrieved context boosts the LLM's confidence, which is used to fine-tune a reranker for more effective context selection; and second, CBDR (Confidence-Based Dynamic Retrieval), a mechanism that adaptively triggers retrieval based on the LLM's initial confidence to avoid unnecessary searches. Experimental results demonstrate that this integrated framework achieves a significant 5.6 percentage point (pp) increase in RAG accuracy while concurrently reducing retrieval costs by 7.1 pp, thereby substantially improving practical utility without compromising performance.

**Strengths:**

1. The paper is well-organised and presented so that it is quite easy for the audience (even for those with limited knowledge of RAG) to follow the paper with a clear blueprint of the proposed method of key components and their functionality in mind.

2. The research is well motivated, and the paper targets the most prominent research problems in the RAG field, i.e., the knowledge boundary problem for efficient and effective RAG.

3. I appreciate the  DISCUSSION section, which cleared many of my questions and helped me get a deeper understanding of the method.

**Weaknesses:**

1. The idea of using a model's latent representation to compute a confidence score is not new, for example:
https://arxiv.org/pdf/2405.18727
2. The general idea of utilising confidence is not a novel idea, making the whole paper less interesting.

**Questions:**

1. Will the reranker be model-specific? Will it require retraining the reranker if the LLM is changed? Retraining the reranker is time-costly, especially for the data gathering process.
2. From the introduction, it seems that the target for this paper is to address the knowledge boundary problem of the RAG system, i.e., mainly the question of ``when to search'', which is addressed by the proposed CBDR mechanism (although similar heuristics with the threshold as the indicator can be widely seen from the existing papers). But to the reviewer, the main contribution is to address the efficiency of the RAG system (addressed by training a preference-based reranker, which is bound to target LLM). Therefore, the core contribution of the paper is to some extent misaligned with the main research question.
3. The general idea of the proposed method is quite similar to the paper below:
https://arxiv.org/pdf/2405.18727

---

> ### Author Response · Authors · 2025-11-21
>
> We appreciate your valuable comments, which have not only helped improve our research but also provided important guidance for subsequent related work. Below, we address each of your questions one by one; please feel free to point out any further issues:
>
> 1. The fine-tuned reranker exhibits strong dependence on the target model—if the target Large Language Model (LLM) is changed, the reranker needs to be retrained. This is because the labels of the reranker’s fine-tuning dataset are directly constructed based on the target LLM’s confidence changes toward different retrieved contexts. We have detailed the correlation between the reranker and the target LLM in the revised manuscript and supplemented a cross-model preference comparison experiment to corroborate this conclusion (see details in Lines 444-455). Regarding training time, we have also conducted a detailed analysis in Lines 509-529 of the manuscript: the data collection phase requires constructing two types of datasets (the training dataset for the confidence detection model E and the reranker fine-tuning dataset). Since the hidden vectors in the datasets are extracted from the Layer/2 layer of the target LLM before generating the first token, only forward passes of the target LLM are required. The entire data collection phase involves only approximately 74,204 forward passes of the target LLM. For the Llama3-8B-Instruct model, the full offline preparation work—including data preparation, model training and fine-tuning, and reading/writing—can be completed within 6 hours on a single NVIDIA RTX 4090 GPU. Moreover, we have extended our evaluation to include related works—specifically DRAGIN and CtrlA—with results reported in Table 3. Specifically, we compare DRAGIN, CtrlA, and our CBDR in terms of: (i) the proportion of queries requiring retrieval under varying confidence thresholds, (ii) final RAG accuracy, and (iii) resource consumption during both offline preparation and online inference. While CBDR incurs higher offline costs than DRAGIN and CtrlA due to confidence modeling, its online inference overhead is negligible. Detailed results and discussion can be found in the main text (see details in Lines 509–529).
>
> 2. Your suggestion on the Introduction section is highly valuable. In essence, the discussion on knowledge boundaries in the Introduction aims to clarify two core research directions currently facing RAG systems: "whether external knowledge is needed" and "which external knowledge is effective." Our research specifically addresses these two directions by enabling the LLM to autonomously decide "when to retrieve" and "which retrieved contexts to prioritize." We have revised the expressions in the Introduction (see details in Lines 35-51), using knowledge boundaries as the background and clearly illustrating how this research specifically improves RAG system efficiency.
>
> 3. We have carefully studied the paper you recommended and included it in the main text citations. As you noted, this work shares similarities with ours in using the model’s internal states to evaluate answer confidence. However, to the best of our knowledge, this is the first study to use "confidence shifts of the model when exposed to different retrieved documents" as a preference signal and apply it to RAG systems to enhance performance. Such confidence shift signals have not been previously used to directly guide the retrieval module (with a focus on the reranker in this study) to return retrieved documents preferred by the target LLM. Therefore, although our research builds on existing work (evaluating confidence using hidden states), it still possesses clear innovation.

---

> > ### Comment · Reviewer_emZB · 2025-11-27
> > **Further comments**
> >
> > I sincerely thank the authors for the detailed explanations regarding the issues I have raised. Part of my concerns have been addressed. However, I still have the following problems that I consider critical.
> >
> > 1. I still think the ideas of using model internal state for better RAG are less novel, given that there have been many papers investigating this point, for example, those I put down in the initial reviews.
> > 2. The re-training of the ranker with respect to different base model limits the scalability of the proposed methods. However, I do understand that other models have different behaviours, thus re-training is needed. Do you have any ways to mitigate such re-training costs if it is compulsory?

---

### Note · Authors · 2025-12-02

**Comment:**

We are withdrawing this submission in order to further polish and refine the paper. We sincerely thank the ICLR reviewers for their insightful and constructive comments, which will greatly help us improve the work and develop a stronger version in the future.

**Withdrawal Confirmation:**

I have read and agree with the venue's withdrawal policy on behalf of myself and my co-authors.